Relationship between walking speed, respiratory muscle strength, and dynamic balance in community-dwelling older people who required long-term care or support and used a daycare center

Jiroumaru Takumi t-jiromaru@bukkyo-u.ac.jp 1
Hyodo Yutaro 2
Wachi Michio 1
Shichiri Nobuko 3
Ochi Junko 1
Fujikawa Takamitsu 1
1 Department of Physical Therapy, School of Health Sciences, Bukkyo University , Kyoto Nakagyo-ku , Kyoto , Japan
2 Department of Rehabilitation, Kanazawa Orthopaedic and Sports Medicine Clinic , Ritto , Shiga , Japan
3 Department of Occupational Therapy, School of Health Sciences, Bukkyo University , Kyoto Nakagyo-ku , Kyoto , Japan
Barak Meir
Electronic publication date: 2023 Dec 21
Publication date: 2023
Volume: 11
Electronic Location ID: e16630
Received 2023 Jul 27; Accepted 2023 Nov 16
Copyright: ©2023 Jiroumaru et al.
Copyright year: 2023
Copyright holder: Jiroumaru et al.
License: This is an open access article distributed under the terms of the Creative Commons Attribution License, which permits unrestricted use, distribution, reproduction and adaptation in any medium and for any purpose provided that it is properly attributed. For attribution, the original author(s), title, publication source (PeerJ) and either DOI or URL of the article must be cited.
License URL: https://creativecommons.org/licenses/by/4.0/

Keywords: Respiratory muscle strength, Walking speed, Dynamic balance, Maximal double step length test, Community-dwelling older people, Required long-term care or support

Funding: The authors received no funding for this work.

==============================
Background

Focusing on the relationship between frail older people and gait speed is vital to minimize the need for long-term care or increased support. The relationship between gait speed, respiratory muscle strength, and dynamic balance, is not well understood in older people requiring long-term care or support. Therefore, this study aimed to provide new insights into the relationship between gait speed, respiratory muscle strength, and dynamic balance in community-dwelling older people who required long-term care or support and used a daycare center.

Methods

This was a cross-sectional study of 49 community-dwelling older people (21 men, 28 women) aged ≥65 years who were certified as requiring long-term care or support under the Japanese system. The participants’ maximal inspiratory pressure (PImax), maximal expiratory pressure (PEmax), walking speed (maximal and normal walking speed), and maximal double-step length test (MDST) results were recorded. The measurement data were evaluated using Pearson’s correlation coefficient and multiple regression analysis.

Results

Pearson’s correlation coefficient revealed correlations between PImax and the following: maximal walking speed (r = 0.606, p < 0.001), normal walking speed (r = 0.487, p < 0.001), and MDST (r = 0.435, p = 0.002). Correlations were also observed between PEmax and the following: maximal walking speed (r = 0.522, p < 0.001), normal walking speed (r = 0.467, p < 0.001), and MDST (r = 0.314, p = 0.028). Moreover, a correlation was found between MDST and both maximal walking speed and (r = 0.684, p < 0.001) and normal walking speed (r = 0.649, p < 0.001). The effect size was 0.379. Multiple regression analysis using a forced entry method with maximal walking speed as the dependent variable showed that maximal walking speed was significantly associated with MDST (p < 0.001) and PEmax (p = 0.036), with an effect size of 0.272. The model’s adjusted coefficient of determination was 0.593 (p < 0.001). Multiple regression analysis using a forced entry method with normal walking speed as the dependent variable showed that normal walking speed was significantly associated with MDST (p < 0.001) and PEmax (p = 0.021), with an effect size of 0.272. The model’s adjusted coefficient of determination was 0.497 (p < 0.001). Multiple regression analysis using a forced entry method with MDST as the dependent variable showed that MDST was significantly associated with PImax (p < 0.025), with an effect size of 0.243. The model’s adjusted coefficient of determination was 0.148 (p = 0.017).

Conclusions

Respiratory muscle strength and dynamic balance were related to walking speed in older people requiring long-term care or support.

Introduction

Japan is currently considered to have a hyper-aged society, and the number of older people who have disabilities, need assistance, or require long-term care has increased rapidly, with a corresponding increase in the economic burden (Ministry of Health, Labour and Welfare, Japan, 2016; United Nations, 2017). Japan’s long-term care insurance system, which began in 2000, is estimated to cost more than $100 billion US dollars, more than three times the cost in 2003 (Ministry of Health, Labour and Welfare, Japan, 2016). Against this background, in a study examining the efficacy of preventive services in Japan’s long-term care insurance system, the group aged ≥ 85 years with a low level of disability showed significantly less deterioration in the level of long-term care or support (Ito et al., 2021). In contrast, the study reported no efficacy in the other age groups (Ito et al., 2021). These results indicate that preventive services provided by the Japanese long-term care insurance system may be effective only for some older people. Therefore, it is necessary to reexamine the physical function interventions that are effective for most older people.

A decline in physical function is a common feature of older age and has significant consequences in terms of quality of life, falls, health care utilization, hospitalization, and mortality (Freedman, Martin & Schoeni, 2002; Gill & Kurland, 2003). Previous studies have examined activities of daily living (ADL) limitations in the last months or years before death (Lunney et al., 2003; Lunney et al., 2018b; Lunney et al., 2018a). Landré et al. (2021) found that older people with a faster gait, higher muscular grip strength, and greater ability to stand up from a chair and who maintain the motor skills to perform daily activities, such as bathing and dressing, may have a lower risk of death compared with older people with reduced motor function. The authors also found that the decline in motor functions begins 4–10 years before death, and that basic/instrumental ADL limitations appear later in life. Regarding gait speed, previous studies have reported a decrease in gait speed as early as 10 years before death (Sabia et al., 2014). Therefore, focusing on the relationship between frail older people and walking speed is essential to minimize the need for long-term care or support.

Recently, the importance of respiratory muscle strength in the physical function of older people has gained attention. Respiratory muscle strength in older people is associated with sarcopenia and frailty (Vidal et al., 2020; Nagano et al., 2021) and may be related to grip strength and walking speed, which are indicators of total body muscle mass (Ohara et al., 2018; Parentoni et al., 2015). Additionally, inspiratory muscle strength is a possible cause of limitations in ADL (Roldán et al., 2021). However, one study reported no correlation between walking speed and respiratory muscle strength in older people (Shin et al., 2017). Notably, the participants in the study were healthy older people with high physical function.

In contrast, a previous study that reported a relationship between walking speed and respiratory muscle strength in older people (Ohara et al., 2018) included participants with sarcopenia and healthy older people. Considering these studies, an association between walking speed and respiratory muscle strength in older people is likely in frail older people. Furthermore, a decrease in walking speed in older people is associated with falls (Kyrdalen et al., 2019) and an increased risk of needing long-term care or support (Ministry of Health, Labour and Welfare, Japan, 2019). However, studies examining the relationship between walking speed and respiratory muscle strength only in frail older people, such as those who require long-term care or support, are lacking.

Walking speed is associated with dynamic stability, and instability increases gradually at slower walking speeds in older people compared with younger people (Kongsuk, Brown & Hurt, 2019). Additionally, a decrease in dynamic stability in older people may lead to falls (Bizovska et al., 2018; Bohm et al., 2020). In Japan, the maximal double-step length test (MDST) has been used as a dynamic balance index to evaluate locomotive syndrome (Japanese Orthopaedic Association, 2015) (a condition in which mobility in daily activities is impaired owing to reduced balance and mobility, muscle weakness, and pain caused by musculoskeletal disorders). However, no studies have examined the relationship between the MDST, respiratory muscle strength, and walking speed exclusively in older people who require long-term care or support.

We hypothesized that walking speed, respiratory muscle strength, and dynamic balance are interrelated in older people who require long-term care or support. A better understanding of this relationship may improve the quality of care services in the long-term care insurance system and reduce the risk of the requirement for long-term care or support. Therefore, this study aimed to investigate the relationship between walking speed, respiratory muscle strength, and dynamic balance among older people living in the community who required long-term care or support and used a daycare center.

Materials and Methods

Study design

This is a cross-sectional study performed at two daycare centers in a medium-sized city in Shiga, Japan, from April 2023 to June 2023. The two daycare centers were close to each other and had similar community environments and care services. All participants were informed of the study details verbally and in writing, and written consent was obtained from each participant. This study was performed in accordance with the guidelines of the Declaration of Helsinki and was approved by the Ethics Committee of the Kanazawa Orthopedic Sports Medicine Clinic (kanazawa-OSMC-2023-003).

Study participants

This study enrolled 49 community-dwelling older people (21 men and 28 women) aged ≥ 65 years who used daycare services in Japan. Participants were certified as requiring long-term care or support under the Japanese system (Yamada & Arai, 2020). The daycare service for older people at the target facilities provided rehabilitation services, including exercise and transportation options. The participants in this study had no cognitive difficulties. Additionally, walking did not require physical assistance, or the participants used walking aids for ADLs. Potential participants with prominent pain or postural abnormalities that would make it difficult to perform the walking speed test or MDST were excluded. Figure 1 is a study flowchart showing the selection of the study participants. Sixty participants (30 men and 30 women) were recruited, of whom, 11 were excluded (seven with difficulty walking or difficulty MDST due to postural abnormalities, two with difficulty walking or difficulty MDST due to knee joint pain, one with difficulty walking or difficulty MDST due to low back pain, and one with extremely low PEmax). The final sample comprised 49 study participants (21 men and 28 women) (Table 1). We also excluded participants with extremely low PEmax. The order in which each test (respiratory muscle strength, gait speed, and MDST) was performed was randomized to avoid bias due to the measurement order.

Respiratory muscle strength

The respiratory muscle strength parameters used in this study were maximal inspiratory pressure (PImax) and maximal expiratory pressure (PEmax). Respiratory muscle strength was measured using a spirometer (Autospiro AS-507; Minato, Japan). All measurements were recorded, and all tests were supervised, by a physical therapist in accordance with the American Thoracic Society/European Respiratory Society guidelines (American Thoracic Society/European Respiratory Society, 2002). PImax and PEmax were used to indicate respiratory muscle strength, and the maximal values after two measurements were considered representative for each parameter.

Walking speed

Walking speed was assessed using a 5 m walking test on a flat floor. The same physical therapist supervised all tests and ensured the tests were performed under safe conditions. The test was practiced by the participants 3–5 times in advance, and participants were asked to wear their usual shoes during the test. Participants were instructed to start from a standing position and walk for 5 m without slowing down in front of the 5 m line. The time from when the first foot passed the starting line until the first foot passed the 5 m line was measured using a handheld stopwatch. This measurement was divided by the distance walked to obtain the average speed (m/s). Participants were asked to perform two walking speed tasks, one at their habitual daily pace (normal walking speed) and the other at their maximal speed (maximal walking speed). Walking at normal and maximal speeds was performed twice each, and the maximal measurement was adopted for the analyses. The tests were performed under the close supervision of a physical therapist to avoid problems such as falls. No physical assistance was provided to the participants in any of the tests. However, participants were allowed to use personal aids (e.g., canes, walkers).

Dynamic balance (MDST)

The maximal double-step length test (MDST) was supervised by the same physical therapist as that for the walking speed evaluations. The test was performed on a flat floor in the daycare facilities under safe conditions. The MDST was practiced by the participants 3–5 times in advance, and participants were asked to wear their usual shoes when performing the test. The MDST was measured twice in the standing position in accordance with the original method, with the participant not losing their balance, and the maximal measurement (cm) was recorded. The MDST was measured from the starting line (toe) to the heel of the second foot, based on the method of Demura & Yamada (2008). The order of the preceding swinging lower limb was arbitrary. It was assumed that the participants in this study would have lower movement ability than those in Demura & Yamada (2008) study. Therefore, the 5-cm measurement used in the previous study was judged problematic for capturing differences in ability among our participants, and a more detailed measurement unit of 0.5 cm was used. Each value for the MDST results was divided by the participant’s height (cm), considering the influence of body composition. The tests were performed under the strict supervision of a physical therapist to avoid problems, such as falls. No physical assistance was provided to the participants in any of the tests. However, participants were allowed to use personal assistive devices (e.g., canes, walkers).

Statistical analyses

The relationships between walking speed, respiratory muscle strength, and the MDST were examined using Pearson’s correlation coefficient and multiple regression analysis. Before analyzing the data, normality of the distribution of the variables was evaluated for maximal walking speed, normal walking speed, PImax, PEmax, and the MDST. The distribution was assessed for each variable using histograms, Q–Q plots, and the Shapiro–Wilk test. The results of the Shapiro–Wilk test was as follows: maximal walking speed: p = 0.468; normal walking speed: p = 0.231; PImax: p = 0.097; PEmax: p = 0.170; and the MDST: p = 0.290. As all p-values were above the conventional significance level of 0.05, we considered all variables statistically normally distributed. With these assumptions confirmed, we proceeded with the subsequent statistical analyses.

Multiple regression analysis was performed using a forced entry method with walking speed (normal and maximal walking speeds) as the dependent variable, and PImax, PEmax, and MDST as independent variables, and sex as adjustment variable, considering multicollinearity. Multiple regression analysis using a forced entry method was also performed with MDST as the dependent variable, and PImax, PEmax as independent variables, and sex as adjustment variable.

As stated, a forced entry method was used for the multiple regression analysis. We calculated the effect size needed for Pearson’s correlation coefficient (sample size = 49, significance level [α] error = 0.05, power [1 − β error probability (err prob)] = 0.80) and multiple regression analysis (sample size = 49, α err prob = 0.05, power [1 − β err prob] = 0.80, number of predictors = 3 or 4) using G Power software (version 3.1; Heinrich Heine University of Düsseldorf, Düsseldorf, Germany). We obtained a result of 0.379 for Pearson’s correlation coefficient and 0.243 or 0.272 for the multiple regression analysis. Statistical analysis was performed using SPSS version 26 (IBM Japan, Tokyo, Japan), and p-values < 0.05 were considered statistically significant.

Results

Pearson’s correlation coefficient indicated a strong positive correlation between maximal walking speed and PImax (r = 0.606, p < 0.001); moderate positive correlation between normal walking speed and PImax (r = 0.487, p <  0.001); and moderate positive correlation between MDST and PImax (r = 0.435, p = 0.002). Additionally, moderate positive correlations were found between maximal walking speed and PEmax (r = 0.522, p <  0.001) and between normal walking speed and PEmax (r = 0.467, p < 0.001). A weak positive correlation was found between MDST and PEmax (r = 0.314, p = 0.028). Moreover, a strong positive correlation was found between maximal walking speed and the MDST (r = 0.684, p < 0.001), and a strong positive correlation was found between normal walking speed and the MDST (r = 0.649, p < 0.001). There was a moderate to strong positive correlation between most of the variables (Table 2).

Figure 1 Flowchart of participant recruitment.

Table 1 Basic attributes and measurement values (means ± SD).

	n = 49 (men = 21, women = 28)	
Basic attributes		
Age	82.6 ± 5.7	
Height	154.1 ± 9.5	
Weight	55.1 ± 10.5	
BMI	23.1 ± 3.3	
Care level	Level 1 = 15, Level 2 = 10, Level 3 = 15, Level 4 = 9, Level 5 = 0, Level 6 = 0, Level 7 = 0	
Measurements values		
Respiratory muscle strength		
PEmax	51.3 ± 20.9	
PImax	36.7 ± 16.6	
Walking speed		
Normal walking speed	0.9 ± 0.3	
Maximal walking speed	1.1 ± 0.4	
Dynamic balance		
MDST	0.9 ± 0.2	
Notes.

Age: years; Height: cm; Weight: cm; BMI (kg/m2), Body mass index. BMI is calculated by dividing the individual’s weight (kg) by the square of their height (m). PEmax (cmH2O), Maximal expiratory pressure; PImax (cmH2O), Maximal inspiratory pressure; Normal walking speed: m/s; Maximal walking speed: m/s; MDST: cm/cm. The maximal double step length test result is calculated by dividing the maximal double step length (cm) by the height (cm).

Table 2 Pearson’s correlation coefficients between the measured variables.

	PEmax	PImax	Maximal walking speed	Normal walking speed	MDST	
PEmax	1.000	0.632***	0.522***	0.467***	0.314*	
PImax	0.632***	1.000	0.606***	0.487***	0.435**	
Maximal walking speed	0.522***	0.606***	1.000	0.931***	0.684***	
Normal walking speed	0.467***	0.487***	0.931***	1.000	0.649***	
MDST	0.314*	0.435**	0.684***	0.649***	1.000	
Notes.

PEmax (cmH2O), Maximal expiratory pressure; PImax (cmH2O), Maximal inspiratory pressure. Maximal walking speed: m/s; Normal walking speed: m/s; MDST: cm/cm. The value for the maximal double step length test is calculated by dividing the maximal double step length (cm) by the height (cm).

* p < 0.05.

** p < 0.01.

*** p < 0.001.

Using multiple regression analysis, with maximal walking speed as the dependent variable, the predictors were PImax, PEmax, MDST, and sex. The resulting model accounted for 59.3% of the variance in maximal walking speed (adjusted R2 = 0.593). Specifically, one standard deviation increase in PEmax was associated with a 0.291 standard deviation increase in maximal walking speed (standard error (SE) = 0.002, β = 0.291, p = 0.036). Similarly, one standard deviation increase in MDST was associated with a 0.291 standard deviation increase in maximal walking speed (SE = 0.160, β = 0.525, p <  0.001). These findings confirm the statistically significant and substantive impacts of PEmax and the MDST on an individual’s maximal walking speed. Using multiple regression analysis, with normal walking speed as the dependent variable, the predictors were PImax, PEmax, MDST, and sex. The resulting model accounted for 49.7% of the variance in normal walking speed (adjusted R2 = 0.497). Specifically, one standard deviation increase in PEmax was associated with a 0.357 standard deviation increase in normal walking speed (SE = 0.002, β = 0.357, p = 0.021). Similarly, one standard deviation increase in MDST was associated with a 0.551 standard deviation increase in normal walking speed (SE = 0.142, β = 0.551, p <  0.001). These findings confirm the statistically significant and substantive impacts of PEmax and the MDST on an individual’s normal walking speed. Using multiple regression analysis and, with MDST as the dependent variable, the predictors were PImax, PEmax, and sex. The resulting model accounted for 14.8% of the variance in the MDST (adjusted R2 = 0.148). Specifically, one standard deviation increase in the PImax was associated with a 0.398 standard deviation increase in the MDST (SE = 0.003, β = 0.398, p = 0.025). This finding confirms the statistically significant and substantive impacts of PImax on an individual’s MDST (Table 3).

Table 3 Multiple regression analysis using a forced entry method, with maximal walking speed, normal walking speed, and MDST as the dependent variables.

Dependent variable	Predictor	β	SE	t-value	p-value	95% confidence interval	Adjusted (R2)	Effect size	
Maximal walking speed	PEmax	0.291	0.002	2.165	<0.036*	[0.000, 0.010]	0.593	0.272	
MDST	0.525	0.160	5.094	<0.001***	[0.492, 1.136]	
Normal walking speed	PEmax	0.357	0.002	2.385	<0.021*	[0.001, 0.009]	0.497	0.272	
MDST	0.551	0.142	4.809	<0.001***	[0.396, 0.966]	
MDST	PImax	0.398	0.003	2.316	<0.025*	[0.001, 0.110]	0.148	0.243	
Notes.

Multiple regression analysis using a forced entry method, with maximal walking speed as the dependent variable indicated that maximal walking speed was significantly associated with MDST and PEmax. Multiple regression analysis using a forced entry method, with normal walking speed as the dependent variable showed that normal walking speed was significantly associated with MDST and PEmax. Multiple regression analysis using a forced entry method, with MDST as the dependent variable showed that MDST was significantly associated with PImax. PEmax (cmH2O), Maximal expiratory pressure; PImax (cmH2O), Maximal inspiratory pressure. Normal walking speed: m/s; Maximal walking speed: m/s; MDST: cm/cm. The value for the maximal double step length test is calculated by dividing the maximal double step length (cm) by the height (cm). β, Standardized coefficient; SE, Standard error.

* p < 0.05.

*** p < 0.001.

Discussion

To our knowledge, this is the first study to examine the relationship between walking speed, respiratory muscle strength, and the MDST, a dynamic balance index, in community-dwelling older people who required long-term care or support and who used a daycare center. This study showed a significant correlation between PImax and maximal walking speed, normal walking speed, and the MDST in the study population. Additionally, significant correlations were found between PEmax and maximal walking speed, normal walking speed, and the MDST and between the MDST and maximal walking speed and normal walking speed.

Moreover, multiple regression analysis showed that MDST and PEmax affected the maximal walking speed of older people who required long-term care or support, and the MDST and PEmax affected the participants’ normal walking speed. Furthermore, PImax was identified as a factor affecting the MDST in older people who required long-term care or support. The results of this study reflect our hypothesis, and the authors consider that the fact that walking speed, respiratory muscle strength, and dynamic balance in older people who required long-term care or support were associated is important for improving the quality of care services in the long-term care insurance system in Japan.

The reason why maximal walking speed was associated with the MDST and PEmax, and normal walking speed with the MDST and PEmax, may involve a breakdown in the postural control system. The postural control system is the process of maintaining posture through the coordination of muscles and the nervous system. This system plays an essential role in maintaining posture and movement stability during walking.

It is widely known that respiratory muscles, which are also trunk muscles, are involved in the postural control system (Hodges & Gandevia, 2000; Hodges & Richardson, 1999; Enright, Unnithan & Heward, 2006; Kocjan et al., 2018). The diaphragm and transversus abdominis are the primary respiratory muscles involved in the postural control system (Hodges & Gandevia, 2000; Hodges & Richardson, 1999). The diaphragm is the major inspiratory muscle (Enright, Unnithan & Heward, 2006). This muscle is thought to play an indirect role in supporting the spine by increasing intra-abdominal pressure as well as a direct role in contributing to postural stabilization through sustained co-contraction (Hodges & Gandevia, 2000; Kocjan et al., 2018). In comparison, the transversus abdominis muscle, one of the expiratory muscles (Tagliabue et al., 2022), is thought to contribute to postural stabilization by increasing the tension of the thoracolumbar fascia (Hodges & Richardson, 1997) or by increasing intra-abdominal pressure (Cresswell, Oddsson & Thorstensson, 1994); thereby, stabilizing the spine (Hodges & Richardson, 1999; Lee et al., 2015).

The strength of the diaphragm and transversus abdominis muscles is likely related to inspiratory and expiratory muscle strength (Souza et al., 2014; Ferraro et al., 2020; McMeeken et al., 2004; Misuri et al., 1997). Therefore, low PImax and PEmax causes a breakdown in the postural control system, affecting maximal and normal walking speeds (Kongsuk, Brown & Hurt, 2019; Yamamoto et al., 2014; Xie et al., 2017). Özkal et al. (2019) noted that the diaphragm is thicker in older people than that in younger people to compensate for atrophied lower limbs and to maintain balance.

Conversely, Noguchi et al. (2021) reported an association between gait speed and the thickness of abdominal muscles, notably the transversus abdominis, in older people requiring long-term care or support. Similarly, the findings of this study indicated that there was a significant relationship between dynamic balance, as measured MDST, and PImax, which implies the potential role of diaphragm functionality in dynamic balance. Moreover, the results of the present study showed that both maximal and normal walking speeds were associated with PEmax, suggesting that the function of the transversus abdominis muscle may be more important than that of the diaphragm for walking speed. Therefore, it is posited that the diaphragm principally contributes to dynamic balance, whereas the transversus abdominis muscle is primarily associated with gait speed.

In summary, the relationships between walking speed, respiratory muscle strength, and dynamic balance in older people who require long-term care or support can be explained through a breakdown of the postural control system due to a decline in respiratory muscle strength. Disruption of the postural control system due to decreased respiratory muscle strength may adversely affect dynamic balance and cause a decrease in walking speed. Walking speed declines approximately 10 years before death, leading to basic/instrumental ADL limitations later in life (Sabia et al., 2014). Because of the decline in walking speed, older people who need long-term care or support may be reluctant to leave home and engage in social activities (Busch et al., 2015; Srithumsuk et al., 2020; Kuys et al., 2014). This leads to social isolation and physical activity limitations, decreasing quality of life and increasing the likelihood of premature death (Studenski et al., 2011; Lyons et al., 2016; Kuang, Huisingh-Scheetz & Miller, 2023).

The major strengths of this study are as follows: First, the study focused on community-dwelling older people who required long-term care or support and evaluated the effects of this requirement on physical function. The findings are very important for Japan’s aging population. This study comprehensively assessed several physical functions, namely, normal walking speed, maximal walking speed, respiratory muscle strength, and dynamic balance. This study also provided a new perspective on how respiratory muscle strength affects walking speed and dynamic balance in older people. In particular, the findings suggest that respiratory muscle strength may play an important role in walking speed and dynamic balance. This finding may contribute to improved care services and quality of life for older people in the long-term care or support insurance system in Japan. Finally, this study evaluated both normal and maximal walking speeds. This evaluation highlighted the continuum from normal daily walking to walking at maximal physical capacity and the importance of respiratory muscle strength for both. In conclusion, this study has many strengths, including a comprehensive assessment of physical function, new insights into the role of respiratory muscle strength, and the use of the MDST. These findings may contribute to improved care services within the long-term care and support insurance system in Japan.

This study also has several limitations. The participants in this study were older people who required long-term care or support, and the participants may have been affected by various medical conditions. However, it was difficult to determine whether the effects on gait speed, respiratory muscle strength, and dynamic balance ability were due to disease. Therefore, additional studies are needed to generalize the results of this study. Future studies should compare sarcopenia and non-sarcopenia groups and frail and non-frail groups, and include healthy older people as a control group. Additionally, we could not evaluate differences between men and women for the evaluated parameters. Future studies should evaluate whether there are differences between men and women in the results of the parameters evaluated in this study. Furthermore, this study was observational, and future interventional studies should be performed. Moreover, the participants in this study were recruited from only two daycare facilities, and it is necessary to continue the study by recruiting participants from multiple centers.

Conclusions

This study revealed that respiratory muscle strength and dynamic balance are associated with walking speed in older people who require long-term care or support. Therefore, improving respiratory muscle strength is recommended as a preventive measure to minimize the need for long-term care and support. The development of such programs and interventions is urgently needed; however, the present study was observational, and the development of these programs requires interventional studies.

Supplemental Information

Supplemental Information 1 Raw data

The relationships between walking speed, respiratory muscle strength, and MDST, were examined using Pearson’s correlation coefficient and multiple regression analysis. In the Multiple regression analysis using a stepwise method, walking speed (normal and maximal walking speed) was defined as the dependent variable, and PImax, PEmax, MDST, sex, height, and weight were defined as independent variables, considering multicollinearity.

Click here for additional data file.

Supplemental Information 2 STROBE checklist

Click here for additional data file.

The authors thank all people who contributed to this study. We thank Jane Charbonneau, DVM, from Edanz for editing a draft of this manuscript.

Additional Information and Declarations

Competing Interests

Author Contributions

Human Ethics

Data Availability

The authors declare there are no competing interests.

Takumi Jiroumaru conceived and designed the experiments, performed the experiments, analyzed the data, prepared figures and/or tables, authored or reviewed drafts of the article, and approved the final draft.

Yutaro Hyodo conceived and designed the experiments, performed the experiments, prepared figures and/or tables, authored or reviewed drafts of the article, and approved the final draft.

Michio Wachi performed the experiments, authored or reviewed drafts of the article, and approved the final draft.

Nobuko Shichiri analyzed the data, authored or reviewed drafts of the article, and approved the final draft.

Junko Ochi analyzed the data, authored or reviewed drafts of the article, and approved the final draft.

Takamitsu Fujikawa analyzed the data, authored or reviewed drafts of the article, and approved the final draft.

The following information was supplied relating to ethical approvals (i.e., approving body and any reference numbers):

The Kanazawa Orthopedic Sports Medicine Clinic granted Ethical approval to carry out the study within its facilities (kanazawa-OSMC-2023-003).

The following information was supplied regarding data availability:

The raw measurements are available in the Supplemental Files.

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
