# Peer review of "Relationship between walking speed, respiratory muscle strength, and dynamic balance in community-dwelling older people who required long-term care or support and used a daycare center"

_PeerJ, doi:10.7717/peerj.16630_

## Round 0.1 · original submission · Major Revisions

Dear Dr. Jiroumaru,

Your manuscript titled " Relationship between walking speed, respiratory muscle strength, and dynamic balance in older people requiring long-term care or support" was considered by two expert reviewers and based on their opinions and my review, the decision is “Major revisions”.

Please carefully read the reviewers’ comments and address them fully in your revised manuscript. In addition, please address the following points:

1. Intro and discussion are a bit long (as indicated also by the reviewers), consider consolidating your text.

2. It was not clear from the manuscript if the authors tested for differences between men and women for the various parameters (beside using sex as a parameter in the multiple regression analysis). I would expect some differences between male/female. Consider giving the data in table 1 for the general sample (n=50) but also per sex (n=21 for men and n=29 for women).

3. Did the authors compare results between the 2 groups of individuals belonging to different daycare centers? Like my previous comment – this is an uncontrolled variable that must be accounted for. Differences in daycare location (temperature, altitude etc.), food, hours of sunlight, level of care by staff, and many more can change the elderly level of activity and the results of this study.

Please ensure that all review, editorial, and staff comments are addressed in a response letter and any edits or clarifications mentioned in the letter are also inserted into the revised manuscript where appropriate.

**Language Note:** The review process has identified that the English language must be improved. PeerJ can provide language editing services - please contact us at copyediting@peerj.com for pricing (be sure to provide your manuscript number and title). Alternatively, you should make your own arrangements to improve the language quality and provide details in your response letter. – PeerJ Staff

·

Basic reporting

First of all, I thank you for the opportunity of reviewing this study.

In general, it is a good study, and it is relevant to the area. I congratulate the authors.

INTRODUCTION: I found the introduction interesting and well-referenced. However, I leave some suggestions in order to make it a little more objective (diminishing the size of the paragraphs and altering the aim a little bit).

METHODS: Well-done. The only thing it was not clear for me was if the MDST has a measurement unit or not.
Other than that, in the statistical analysis, I suggest adding a sentence about the descriptive analyses of the data, if they are normal o non-normal...

DISCUSSION:
In general, I found it was a good discussion. However, I suggest diminishing the size of it, leaving the ideas a little more objective.
I suggest to diminish the size of the paragraphs, thus, I had changed some of them, creating new paragraphs, as identified in the word document.

I also suggest adding the strength of the study before the limitations.

There are some sentences that need to be referenced.

CONCLUSION:

I suggest to change the manner the study is concluded. Once it is an observational study, there is no effect-cause. Then, you can say that preventive services are recommended, but not that respiratory muscle strength will improve walking speed, because the relationship found is not of effect-cause.

TABLES:
Both tables, need a subtitle describing BMI, cm, kg, BMI, Pemax… etc

Experimental design

METHODS: Well-done. The only thing it was not clear for me was if the MDST has a measurement unit or not.
Other than that, in the statistical analysis, I suggest adding a sentence about the descriptive analyses of the data, if they are normal o non-normal...

Validity of the findings

I suggest to change the manner the study is concluded. Once it is an observational study, there is no effect-cause. Then, you can say that preventive services are recommended, but not that respiratory muscle strength will improve walking speed, because the relationship found is not of effect-cause.

Additional comments

First of all, I thank you for the opportunity of reviewing this study.

In general, it is a good study, and it is relevant to the area. I congratulate the authors.

·

Basic reporting

Congratulations to the authors for the study and thank you for the opportunity to review this article.
The comments I will make will be with the aim of improving your work for future publication in this journal.
Line 03: In the title, I suggest making it clear that the elderly evaluated are from the community and that they use a daycare center.
Introduction:
In my opinion, the introduction was a little long, so to reduce the amount of information in this section, I believe that lines 25-28 are not essential at this point.
Replace “However” (line 28) with another word to start the sentence properly.
The textual cohesion and coherence of lines 48-50 was strange. I suggest that the authors try to improve the writing of this passage, for example the expression “Respiratory muscle strength” that is repeated twice in a short space.
The excerpt “Furthermore, a decrease in walking speed in older people is associated with falls [16] and an increased risk of needing long-term care or support (line 52-53) would be better after the authors present the discussion of walking speed. march, which happens in the previous paragraphs.
Materials and Methods
Line 82: Change “This was” to “This is a cross-sectional”.
Line 82: I missed a better specification of where the two daycare centers are. Are they in large urban centers or in small towns?
Lines 97-98: When citing “prior studies”, I advise adding more than one reference.
Line 98: I doubt the authors' decision to exclude patients with pain that could interfere with the gait speed test. Does this interference refer to the impossibility of carrying out the walk? If so, make this information clear in the text.
Results:
Line 164: In table 1 there is a variable called care level (1-7), how this classification is done. I don't see the description of this in the methodology.
Line 165: I suggest presenting such data in a table and in the text telling us whether the correlations were weak, moderate, or strong, positive or not.
Lines 174, 178 and 182: I suggest adding the beta (regression result) and meaning of this finding.
I don’t have suggestion for discussion and conclusion.

Experimental design

no comment

Validity of the findings

no comment

Additional comments

no comment

Reviewer 3 ·

Basic reporting

Unfortunately, this is not the kind of content that will be published in peerJ.

English needs to be improved.

In the raw data, BMI calculation error, participants with PEmax of "6.9" should be excluded.
If the correlation between multicollinear items is to be described, all of them should be presented in a table for clarity.

Experimental design

There is no novelty beyond the previous studies.

There is a serious problem with this paper. Walking speed and MDST are similar to each other. The multivariate model with walking speed as the dependent variable and MDST as the independent variable is inappropriate.

The number of participants is too small.

The flowchart is mismatched with the text. Although the start of the flowchart is for those who were able to measure their body composition, there is no mention of body composition being measured in the text.

Validity of the findings

The discussion and conclusions are obviously too strong.

Additional comments

There is a mismatch between the measures described in the human participant information and those presented in the paper.

---

## Round 0.2 · Minor Revisions

Dear Dr. Jiroumaru,

Your manuscript titled " Relationship between walking speed, respiratory muscle strength, and dynamic balance in community-dwelling older people who required long-term care or support and used a daycare center " was considered by three expert reviewers and based on their opinions and my review, the decision is “minor revisions”.

Please carefully read the reviewers’ comments and address them fully in your revised manuscript. In particular, please refer to reviewer #3 concerns, specifically - the use of multivariate analysis and the mismatch between the text and the flowchart.

Please ensure that all the reviewers comments are addressed in a response letter and any edits or clarifications mentioned in the letter are also inserted into the revised manuscript where appropriate.

Please note that submitting a revision of your manuscript does not guarantee eventual acceptance, and that your revision may be subject to re-review by the reviewer(s) before a decision is rendered.

·

Basic reporting

I thank the authors for having worked hard to answer my considerations. Thus, I suggest to accept the paper for publication.

Experimental design

My considerations had been answered.

Validity of the findings

All my considerations had been answered.

Additional comments

The authors worked hard to improve the paper. Thus, I suggest to accept the article for publication.

·

Basic reporting

Congratulations to the authors for the significant improvement in their work. All the suggestions I made in the first reading were corrected.

Experimental design

no comment

Validity of the findings

no comment

Additional comments

no comment

Reviewer 3 ·

Basic reporting

The conclusion is too much of a leap. We only know the relationship between gait, respiration, and MDST, but we have no idea about the effect on death, ADL, etc.

Experimental design

The multivariate analysis of this study is inappropriate based on the following.

1. The number of people is overwhelmingly small.
2. Respiration is clearly a gender-differentiated function, so it must be done separately for men and women or gender must be included in the adjustment variable.
3. In the design of this study, if the dependent variable is walking speed, PEmax, PImax, and gender should be entered as independent variables. MDST should not be included here. This is because walking and MDST are conceptually similar, and the regression equation will be significantly oriented.
4. Multivariate analysis should include 95%CI.

Also, extremely low PEmax (6.9) should be excluded.

As I pointed out during the initial peer review, the excluded persons in the flowchart and the text still do not match. For example, we have no idea where in the follow chart those with a forced expiratotry volume in 1 second of<70%

Validity of the findings

No findings that go beyond previous previous studies.

---

## Round 0.3 · accepted · Accept

Dear Dr. Jiroumaru,

Thank you for submitting your revised manuscript titled "Relationship between walking speed, respiratory muscle strength, and dynamic balance in community-dwelling older people who required long-term care or support and used a daycare center". After reading the revised manuscript and the reviewer’s comments (see attached) I’m happy to let you know that decision is “accept”.

I have a few very minor suggestions that should not take more than a few minutes to implement and that would improve the readability of the manuscript.

- L142-4: please move the first results paragraph (L218-23) after these lines. The number of excluded individuals fits here (materials and methods section) and is not part of the results.

- L291: The sentence (Ozkal...) should continue the previous paragraph and not start a new one.

- Table 1: this is just a suggestion but the footnotes for this table are very long. A big part of the footnote is dedicated to describing the 7 care levels. I assume these are documented somewhere else before (paper, official document etc.). Since “care level” is not mentioned anywhere else in the manuscript, I believe a reference is a better option than listing the whole range.

Good luck and thank you for your hard work.
Meir Barak